# How Fermentation Affects the Antioxidant Properties of Cereals and Legumes

**DOI:** 10.3390/foods8090362

**Published:** 2019-08-24

**Authors:** Michela Verni, Vito Verardo, Carlo Giuseppe Rizzello

**Affiliations:** 1Department of Soil, Plant and Food Science, University of Bari Aldo Moro, 70126 Bari, Italy; 2Department of Nutrition and Food Science, University of Granada, Campus Universitario de Cartuja, E-18071 Granada, Spain; 3Institute of Nutrition and Food Technology ‘José Mataix’, Biomedical Research Centre, University of Granada, Avenida del Conocimiento s/n, E-18071 Granada, Spain

**Keywords:** Lactic acid bacteria, fungi, phenolic compounds, bioactive peptides, grains

## Abstract

The major role of antioxidant compounds in preserving food shelf life, as well as providing health promoting benefits, combined with the increasing concern towards synthetic antioxidants, has led the scientific community to focus on natural antioxidants present in food matrices or resulting from microbial metabolism during fermentation. This review aims at providing a comprehensive overview of the effect of fermentation on the antioxidant compounds of vegetables, with emphasis on cereals- and legumes- derived foods. Polyphenols are the main natural antioxidants in food. However, they are often bound to cell wall, glycosylated, or in polymeric forms, which affect their bioaccessibility, yet several metabolic activities are involved in their release or conversion in more active forms. In some cases, the antioxidant properties in vitro, were also confirmed during in vivo studies. Similarly, bioactive peptides resulted from bacterial and fungal proteolysis, were also found to have ex vivo protective effect against oxidation. Fermentation also influenced the bioaccessibility of other compounds, such as vitamins and exopolysaccharides, enabling a further improvement of antioxidant activity in vitro and in vivo. The ability of fermentation to improve food antioxidant properties strictly relies on the metabolic activities of the starter used, and to further demonstrate its potential, more in vivo studies should be carried out.

## 1. Antioxidant Compounds in Food Matrices

The increasing interest towards healthier food and lifestyles has steered the scientific community to pay great attention to the field of free radicals and antioxidant compounds. Free radicals are atoms, molecules, or ions containing an unpaired electron, which makes them unstable and highly reactive [1]. The generation of free radicals in our body occurs as consequence of exposure to different physiochemical conditions or pathological states. Free radicals are responsible for damaging biologically relevant molecules and lipids, nucleic acids, and proteins are the major targets [2]. They are generated continuously; however, they are also involved in inter-cellular and intra-cellular signaling systems and enzymatic reactions essential to intermediary metabolic processes of life [3], therefore, their daily production must be balanced. If free radicals overcome the body’s ability to regulate them, and the ratio between free radical generation and antioxidant defenses is unbalanced, a condition known as oxidative stress ensues. Being the oxidative stress responsible for an enormous number of conditions, including cancer, cardiovascular and neurodegenerative diseases, atherosclerosis, inflammatory state, and many others [2,4], preventing it by enhancing the intake of dietary antioxidants represents the most feasible way of protection against free radicals. Antioxidants need to be capable of delaying or inhibiting the oxidation of a substrate, yet an important property they should also have, is the ability to form a new radical that is stable through intramolecular hydrogen bonding on further oxidation [1]. Several mechanisms of action can be responsible for their activity. Antioxidants can (i) scavenge species that initiate peroxidation, (ii) donate a hydrogen or an electron, (iii) chelate metal ions preventing the generation of reactive species or lipid peroxides de-composition, (iv) quench the radical O_2_^−^ preventing peroxides formation, (v) breaking autoxidative chain reaction, (vi) inhibit pro-oxidative enzymes, and/or (vii) reduce localized O_2_ concentrations [1,5]. From a technological point of view, antioxidants are designed to prevent food from spoiling through oxidation, thus reducing loss of nutrients, and maintaining texture, color pigments, taste, freshness, functionality, and aroma [3]. Therefore, antioxidants are an important category of food preservatives and can be divided into natural or synthetic. Natural antioxidants include flavonoids, phenolic acids, carotenoids, and tocopherols [1]; however, other protein derived compounds, such as amino acids and bioactive peptides, have received great attention for their displayed antioxidant properties [6,7,8]. Both natural and synthetic antioxidants, act by similar mechanisms and their chemical structure and polarity influence the antioxidant activity [5]. Antioxidants efficiency generally increases with the increase of their concentration; however, the dependence is not linear, and when the maximum activity is reached, it may also decrease [9]. Except for carotenes, tocopherols, and their esters, natural antioxidants are mostly much more polar than synthetic ones. They are also less active and substrate specific, and their antioxidant activity depends highly on synergists factors [9].

This review aims at providing a comprehensive overview of the effect of fermentation on the antioxidant activity of vegetable matrices. The literature is full of articles claiming the potential of food fermentation in improving antioxidant properties; however, the method used to quantify phenolic compounds, unless performed by chromatographic analysis is the classic Folin-Ciocalteu, which suffers from several drawbacks. The test is sensitive to pH, temperature, and reaction time; inorganic and non-phenolic organic substances, including reducing agents, react with Folin reagent, causing overestimations of the phenolic content [10,11]. For this reason, all the examples reported in this review were chosen among those in which a characterization of the phenolic profile was performed. Moreover, among vegetable matrices, only grains were considered since (i) diets in developing countries are primarily based on cereals and legumes, whereas in Western society there is an increasing interest in strictly vegetarian diets [12]; (ii) cereals are often consumed as refined products, yet most of the bioactive compounds are in the outer layers of the grains, and get lost during milling [13,14]; (iii) legumes consumption is often limited by the presence of antinutritional compounds which fermentation has proven to diminish [13,14]. After an extensive research on multiple databases, 76 research articles, half of which published in the last five years, highlighting the effect of fermentation on grains antioxidant properties were selected for this review. Respectively, 39 and 17 papers reported phenolic and proteic compounds as responsible for the antioxidant activity. Moreover, such activity was demonstrated in vivo in about 15% of the studies considered.

### 1.1. Phenolic Compounds

The antioxidant activity of phenolic compounds lies in their ideal chemical structure, facilitating the hydrogen or electron donation from hydroxyl groups positioned along the aromatic ring and conferring radical scavenging activities and metal-chelating potential. Phenolics have the ability of stabilizing and delocalizing the un-paired electron within their aromatic rings [15]. Phenolics are composed of at least one aromatic ring with at least one hydroxyl group and may be classified based on the number of phenol rings and the structural elements that are bound to the rings [16]. Phenolic acids, flavonoids, tannins, stilbenes, and lignans are the main groups of phenolics.

Phenolic acids are divided into hydroxybenzoic acids and hydroxycinnamic acids and usually act as antioxidants by trapping free radicals. Flavonoids, instead, can scavenge free radicals and chelate metals [5,15]. The common characteristic of flavonoids is the basic 15-carbon flavan structure. They are arranged in three rings (A, B, and C) and the different classes vary for the level of saturation of the C ring, whereas compounds within the same class differ for the level of substitution of A and B rings [5]. Polyphenols stability and free radical scavenging potential depend on both the number and location of the free OH group. The antioxidant activity increases with the increase of the hydroxyl groups, especially if positioned in ortho-3,4 [17].

Based on their chemical structure, tannins are defined as hydrolysable or condensed (proanthocyanidins). Condensed tannins are oligomers and polymers of flavan-3-ols, whereas hydrolysable tannins are glycosylated gallic and ellagic acid derivatives [18]. It was proven that free radicals scavenging activity of both hydrolysable and condensed tannins involves a fast and a slow step [19]. The fast scavenging reaction is inhibited by complexation of the tannin with protein, a very tight bond between the phenolic group of tannins and the NH group of proteins, which prevents their hydrolysis and digestion in the stomach [19]. Nevertheless, the overall capacity of the tannin-protein complex for scavenging seems to be similar to that of the free tannin [18,19].

### 1.2. Antioxidant Peptides and Protein Derivatives 

Besides their nutritional, physicochemical, and sensory properties, proteins can be responsible for health promoting benefits, mainly attributed to biologically active peptides [20]. Bioactive peptides are produced by digestive enzymes during gastrointestinal digestion, or by proteolytic enzymes during food processing (ripening, fermentation, cooking), storage, or in vitro hydrolysis [7]. These peptides may play various roles (antimicrobial, antihypertensive, hypocholesterolemic, immunomodulatory, antioxidative, antithrombotic, antitumoral) and can be released from native proteins that derive from vegetable or animal matrices [7,8].

Peptides displaying antioxidant activity usually contain 5–16 amino acid residues [21]. The exact mechanism behind it has not fully been understood, yet several studies reported that they are inhibitors of lipid peroxidation, scavengers of free radicals and chelators of transition metal ions. [8,21]. Tyrosine, tryptophan, methionine, lysine, cysteine, and histidine are examples of amino acids displaying antioxidant activity. Synthesized peptides containing the active fragments have been proven to inhibit lipid peroxidation, while the tripeptides, Tyr-HisTyr, and Pro-His-His were found to be effective in stabilizing radical and non-radical oxygen species, including peroxynitrite and lipid peroxide [22]. Amino acids with aromatic residues can donate protons to electron deficient radicals, whereas histidine-containing peptides, thanks to the imidazole group, have been found to have hydrogen-donating, lipid peroxyl radical trapping, and metal ion-chelating abilities [8]. On the other hand, sulphur containing amino acids, have antioxidant action due to the direct interaction of thiol group with radicals. Cysteine and homocysteine inhibit LDL (low-density lipoprotein) oxidation by hemin and copper, methionine residues, instead, scavenge oxidizing agents [6]. Besides the presence of the proper amino acids, their specific positioning in the sequence plays an important role in antioxidant activity of peptides as well as other factors, such as the structure, amino acids configuration, hydrophobicity, and concentration [8]. 

### 1.3. Synthetic Antioxidants 

Synthetic antioxidants were developed to prolong food shelf life but also because of the need to have a standard measurement system to compare with natural antioxidants. There are numerous compounds used in food, animal, and cosmetic applications to prevent oxidation; some also have antifungal properties and possess at least one phenolic ring in the structure. Butylated hydroxyanisole (BHA), butylated hydroxytoluene (BHT), tert-butylhydroquinone, propyl gallate, octyl gallate, 2,4,5-Trihydroxy butyrophenone, nordihydroguaiaretic acid, and 4-Hexylresorcinol are some examples [1]. Among these, BHT and BHA, alone or in combination with others, are the most commonly employed [2] and synergistic effects were also reported [23]. Today, almost all processed foods contain synthetic antioxidants and, despite being reported safe in the past, several studies have addressed their danger for human health, among which liver, kidney, and lungs damages, mutagenesis, carcinogenesis, and many others [19]. For this reason, between 2011 and 2012, the European food safety authority re-evaluated their maximum levels of intake in adults and children [24,25]. Considering the increasing risk factors related to synthetic antioxidants, there has been a global trend toward the use and the search for effective natural substance as therapeutic antioxidants.

## 2. Bioaccessibility and Bioavailability of Antioxidant Compounds

The research around antioxidants has grown exponentially, but there are still limitations that need to be considered before the real potential of these molecules is properly appreciated. All the bioactive compounds, to exert their biological properties, must be available in the target tissue, which is why, when addressing a specific property, it is important to consider their absorption in the gastro-intestinal tract. The absorption depends on the compound bioaccessibility and bioavailability. Only polyphenols that reach the gut, released from the food matrix by the action of digestive enzymes and gut microbiota, are bioaccessible, and therefore, potentially bioavailable [26]. 

Among polyphenol classes, their physicochemical characteristics play a significant role in the overall availability. It was observed that the absorption of phenolic acids, having small-molecular weight, as well as flavones, catechins, and quercetin glucosides results easier compared to larger polyphenols such as proanthocyanidins, which need to be degraded into monomer or dimer units before being absorbed [27,28]. Anthocyanins can be absorbed as glycosides and appear as such in blood [29], whereas galloylated monomeric flavonols do not seem to undergo extensive metabolism [30]. Another important factor to consider when assessing polyphenols bioavailability is their metabolism and the biotransformation reactions they can undergo once absorbed. As a result of the changes in their structure they may or may not still exert the biological action [28].

As for peptides, their ability to resist enzymatic digestion in the gastrointestinal tract is necessary to ensure their bioactivity within the human body. Since peptides’ potential properties strictly depend on the amino acid composition, the loss of amino acid residues as a result of proteolysis in the gastrointestinal tract can lead to changes in the activity. Although it is more likely that small molecules get absorbed without undergoing further digestion, studies have showed that peptides with higher molecular weight can pass to the plasma without modifications and it seems that the amino acids composition is a factor of key importance in the resistance during the digestion tract [31].

Even though the use of in vitro tests has generated a controversy over the last years, these methods are still of great importance in the selection of potential antioxidant compounds. Therefore, in vitro assays are still necessary to screen among the thesis, yet major effort should be put into validate in vivo the bioaccessibility of such compounds. Despite the high reliability recognized to the in vivo studies, the problems related to the approval for the ethical committee, the long times and the high costs and the dependence on the individual response, pushed the development of in vitro methods simulating human digestion. The most widely used procedure for screening bioaccessibility is the in vitro static gastro-intestinal method. However, this method does not include a colonic phase where compounds may be metabolized by the colonic microbiota. A more reliable assessment can be obtained with dynamic gastro-intestinal models, which include the biological environment of the intestine [26].

## 3. Estimation of the Antioxidant Activity in Foods

### 3.1. In Vitro Assays

Although dozens of methods exist to investigate antioxidant activity, only a few are commonly employed to assess food antioxidant potential (for a review see Alam et al. [4]). Among scavenging activity assays, ABTS (2,20-azino-bis (3-ethylbenzothiazoline-6-sulfonic acid)) and DPPH (2,2-diphenyl-1-picrylhydrazyl) are colorimetric assays where the radicals decolorize in the presence of antioxidants [1]. In the hydroxyl radical scavenging, activity fluorescein is used as a probe and the fluorescence decay curve is monitored in the presence and absence of the antioxidants. ORAC (oxygen radical absorbance capacity) is a test performed using Trolox (a water-soluble analog of Vitamin E) as a standard and is based on the generation of free radical and measurement of decrease in fluorescence in the presence of free radical scavengers [4]. Another assay commonly used is the ferric reducing-antioxidant power (FRAP), characterized by the reduction of Fe^3+^ to Fe^2+^ followed by the alteration of color from yellow to blue and analyzed through a spectrophotometer [4]. Several other assays, among which thiobarbituric reactive substances (TBARS) and glutathione peroxidase (GSHPx) methods, evaluate the inhibition of lipid peroxidation [1]. Considering the limitations of in vitro methods: (i) presence in the extracts of pigments and fluorophores interfering with absorbance and fluorescence readings, (ii) failure to evaluate radical scavenging rate, and (iii) lack of biological relevance due to the use of artificial radicals not found in food or biological systems; antioxidant activity should not be concluded based on a single antioxidant test model [11].

### 3.2. Ex Vivo Assays

Since in vitro assays fail to predict the antioxidant activity in vivo and testing a substance directly on animals or human is not an easy approach, methods comprising cellular models for a rapid initial screening have been developed. The hemolysis inhibition assay includes the use of plasma as substrate of oxidation. When exposed to ROS (reactive oxygen species), the oxidation of protein (hemoglobin) and lipid (mainly cholesteryl ester) begins, leading to destruction of the cell shape and membrane structure and ultimately hemolysis. The degree of hemolysis is determined spectrophotometrically measuring the concentration of released hemoglobin in the solution and the inhibition of hemolysis by antioxidants calculated by comparing with a control containing no antioxidants [11]. In cell culture models, human or animal cell lines such as keratinocyte or fibroblasts are subjected to oxidative-induced stress and then incubated for a certain amount of time with the substance to test. At the end of incubation, MTT (3-(4,5-dimethylthiazol-2-yl)-2,5- diphenyltetrazolium bromide) assay, which evaluates the ability of succinate dehydrogenase to convert MTT into formazan crystals in viable cells is performed [32,33]. Otherwise, in cellular antioxidant assays, a fluorescent probe is introduced into the cell cultures and in the presence of ROS or RNS (reactive nitrogen species), the substance is excited emitting fluorescence. The fluorescence intensity measured is proportional to the level of oxidation. Antioxidants absorbed into the cells scavenge the radicals, resulting in lower degree of oxidation observed as attenuated fluorescence increase [11]. Besides cellular assays, for the ex vivo evaluation of antioxidant activity, other biological systems exist, such as the inhibition of LDL and DNA oxidation assays. In the LDL-cholesterol assay, oxidation is induced by transition metals or peroxyl radical, and LDL is incubated with the samples. Then, the extent of the oxidation is determined by measuring the generated amount of lipid peroxides and by the TBARS assay [4]. DNA oxidation assay is based on a similar principle, but in this case, the DNA strand breaking is induced by hydroxyl or peroxyl radicals because they are the major sources responsible for DNA oxidative damage, especially mitochondrial [11].

### 3.3. In Vivo Assays 

In vivo protocols commonly include the administration of antioxidants to testing animals for a specified period of time, after which the animals are sacrificed, and blood or tissues analyzed [4]. The lipid peroxidation (LPO) assay, which measures spectrophotometrically the end products of LPO process in the tested tissue, is one of the most used. GSHPx, instead, catalyzes the reduction of hydroperoxides. GSHPx measurement is performed especially in patients who are under oxidative stress; low activity of this enzyme is one of the early consequences of a disturbance of the prooxidant/antioxidant balance. While the FRAP assay, which was originally applied to plasma, is one of the most rapid tests and very useful for routine analysis [4].

## 4. Effect of Microbial Fermentation on the Antioxidant Activity

### 4.1. Metabolic Activities Affecting Phenolics

Plant phenolics are known to possess antimicrobial properties against bacteria, fungi, and yeasts; therefore, the ability to metabolize them comes from the need of detoxifying such compounds that, if present at high concentrations negatively affect the integrity of the cell wall and membrane, dissipate the pH gradient, delay the metabolism of carbohydrates and denature proteins [34,35,36,37]. Whether it is carried out by fungi, yeasts, or bacteria, microbial fermentation has an impact on the phenolic compounds characterizing food matrices and metabolic activities are species- or strains-specific and depend on their portfolio of enzymes. A schematization of the major effects of fermentation on phenolic compounds is reported in Table 1. 

#### 4.1.1. Metabolic Activities Affecting Phenolic Acids

Phenolic acids are by far the most important food phenolics in terms of quantity, they represent one-third of dietary phenolics and can be present in soluble form within the cytoplasm or bounded to the cell wall [69]. Hydroxybenzoic and hydroxycinnamic acids may be decarboxylated by lactic acid bacteria (LAB) to the corresponding phenol or vinyl derivatives or hydrogenated by phenolic acid reductases [70]. Metabolites of phenolic acids conversion, compared to their precursors, have reduced antimicrobial activity [36] and it was also hypothesized that LAB use hydroxycinnamic acids as external acceptors of electrons, which allow them to gain one extra mole of ATP [71,72]. Strains of *Lactobacillus rossiae*, *Lactobacillus brevis*, and *Lactobacillus curvatus* have followed one of the two paths (decarboxylation or reduction), whereas strains of *Leuconostoc mesenteroides* and *Lactobacillus fermentum* were found not capable of metabolizing hydroxycinnamic acids [71]. Phenolic acids metabolism is influenced by the composition and intrinsic factors of the matrices, therefore, depending on the substrate, the metabolism can shift from decarboxylase to reductase [72]; nevertheless, the derivatives exert higher biological activities than their precursors [73].

Phenolic acid reductase and phenolic acid decarboxylase activities contributed to polyphenol metabolism in red sorghum fermented with *Lactobacillus plantarum* and *Lb. fermentum*. Ferulic acid was reduced to dihydroferulic acid, and caffeic acid was metabolized to vinylcatechol and ethylcatechol but also dihydrocaffeic acid [49]. Similar results were obtained during the fermentation of a malt-based beverage with a pool of LAB including *Lb. plantarum*, *Lb. brevis*, and *Lactobacillus amylolyticus* [51]. Savolainen et al. [40] studied the role of oxygen during the fermentation of a liquid wheat bran sourdough. It was observed that anaerobic conditions, in which lactic acid bacteria and endogenous heterotrophic bacteria grew better, induced the conversion of ferulic and caffeic acids into their corresponding derivatives, and increased the amount of sinapic acid. Aerobic conditions, which favored yeasts growth, was characterized by the presence of dihydroxyphenyl ethanol and hydroxyphenylacetaldehyde.

Phenolic acid metabolism was also reported in yeasts and fungi. Cinnamate caboxy-lyase activity, which transforms coumaric and ferulic acids into their vinyl derivatives, was reported in *S. cerevisiae* strains [74]. Species belonging to *Aspergillus*, *Fusarium*, and *Pycnoporus* genera as well as *Pseudomonas* were responsible for the decarboxylation of ferulic acid and eugenol and their further metabolization, through a lyase, to vanillin, vanillic, and protocatechuic acids [75,76].

As mentioned above, phenolic acids are often bound, as dimers, trimers and/or oligomers, to the plant cell wall polysaccharides such as xylan and pectin. Another class of enzymes involved in their metabolism is represented by feruloyl esterases which are capable of releasing ferulic acid and other cinnamic acid. Feruloyl esterases have been described in lactic acid bacteria, mostly *Lactobacillus plantarum* strains [69,77], in fungi of the genus *Aspergillus* and *Penicillium* [78,79], as well as in some *Bacillus*, *Pseudomonas*, and *Pseudoalteromonas* strains [77,79,80], and it was also hypothesized in *S. cerevisiae* [81]. Several authors studied the impact of bioprocessing with baker’s yeast, LAB, and fungi, with or without the addition of commercial enzymes, on phenolic acids release in wheat bran [39,40,41,81], rye bran [46,47,48], rice bran [43,44], sorghum [49], and tef [51]. In all cases, substantial increases, especially of ferulic acid, were observed. Ferulic acid antioxidant properties are ascribed to its ability to inhibit lipid peroxidation and LDL oxidation greater than other hydroxycinnamic acids [82]. Anson et al. [38] fermented wheat bran with baker’s yeast and used it to produce a fortified bread having ferulic, *p*-coumaric, and sinapic acids content up to three-fold higher than unprocessed bran. Breads were subjected to gastro-intestinal digestion in vitro, although phenolic compounds bioavailability substantial increased, most of them were recovered from the jejunal compartment; only a small part of them was further metabolized in the colon section. Slightly different results were obtained by Koistinen et al. [83], who fermented rye bran with baker’s yeast. Despite the extensive phenolic acid release caused by the bioprocessing (up to 30-fold higher than unfermented sample), when subjected to in vitro colon model, no differences were observed among the thesis.

Among fermented legumes, phenolic acid decarboxylase and esterase activities were reported in fermented cowpeas [66], lentils [67,68], and chickpea [84]. Soy phenolic composition, on the other hand, has been extensively characterized [85], as well as its changes during fermentation with bacterial [55,59,60,61] and fungal strains [54,56,58,86]. Dueñas et al. [54] observed that despite hydroxycinnamic acids content was higher than hydroxybenzoic in unfermented soy flour, hydroxybenzoic acids significantly increased during fermentation, up to seven-fold when *Aspergillus oryzae* was used. On the other hand, *p*-hydroxyphenylacetic acid, which was not detected in raw flours, reached up to 30 μg/g after fermentation [54]. Riciputi et al. [62] also characterized the bound phenolic profile of soymilk fermented with *Lactobacillus casei* and *Lb. acidophilus* to prepare a fermented version of tofu. Bound phenolics were mostly phenolic acids, of which syringic represented more than 30%, followed by *p*-coumaric, ferulic, and *p*-cumaroyl-hexose derivatives, which all increased compared to soybean flour and traditional tofu [62].

#### 4.1.2. Metabolic Activities Affecting Flavonoids

Flavonoids, the other big group of food phenolics, are often glycosylated and several enzymes, belonging to the class of hydrolases, are produced by a great number of microorganisms, both bacteria and fungi [57]. Glycosyl hydrolases convert flavonoid glycosides to the corresponding aglycones, which show higher bioactivity in humans than their precursor glycosides [60]. Glucosidase activity from *Lactobacillus* spp. was responsible for the reduction of flavonoids glycosides in red sorghum, which corresponded to the increase of the aglycones, taxifolin, eriodictyol, and naringenin [49]. Bhanja et al. [87] suggested that besides β-glucosidases, different carbohydrate cleaving enzymes among which amylase and xylanase are responsible for the release of phenolics during solid state fermentation of wheat by *A. oryzae* and *Aspergillus awamori*. Among flavonoids, soy isoflavones are the most extensively studied for their health benefits. As for most flavonoids, they normally occur as glucoside-bound moieties, yet it is the aglycone form that is metabolically active, showing higher antioxidant activity and being absorbed in the intestines faster than their glucoside bound forms [85]. Fermentation with *Lb. casei* and *Lb. acidophilus* enabled the increased of the aglycones/glycosylated ratio in fermented tofu more than 10- and five-fold higher compared to soybean flour and traditional tofu, respectively. Fermentation also increased the content of genistein and daidzein, the major isoflavones found in soybean [62]. Di Cagno et al. [65] fermented soymilk with a pool of LAB strains selected for the high β-glucosidase activity, registering the increase of daidzein, genistein, glycitein, and especially equol. β-glucosidase is the major enzyme involved in isoflavones release and its activity was responsible for the improvement of the scavenging activity on DPPH radical in many soy-products fermented by *Bacillus* spp. [55,85], *Aspergillus* spp. [54,56], *Rhizopus* spp. [54,86], *Lactobacillus* spp. [60,61,63], and *Lentinus edodes* [58]. Dueñas et al. [54] reported that fungal fermentation acted more intensively in releasing isoflavones aglycones compared to flavanones and flavonols. In addition to the β-glucosidase activity, McCue et al. [58] also ascribed the increase of phenolic compounds to a laccase, an enzyme involved in lignin biodegradation by white-rot fungi, suggesting the possibility of a direct activity on polymeric phenolic substrate. In a recent study, instead, the effect of fermentation on soy and chickpea isoflavones was evaluated after legumes seeds were germinated [63]. The authors evaluated the effect of different process parameters adopted during germination. An increase in isoflavones glycosides and aglycones was already observed on sprouts, however, after incubation with *Lb. casei*, especially when seeds were germinated with blue light, a substantial boost in their content was observed.

#### 4.1.3. Metabolic Activities Affecting Tannins

Tannase, which specifically breaks the galloyl ester bonds of hydrolyzable tannins, was first reported for several fungal species of the genus *Aspergillus*. Over the last two decades, many bacterial species of the genera, including *Streptococcus*, *Lonepinella*, *Bacillus*, and *Lactobacillus*, have also been reported to possess tannase activity [88]. However, tannase acting on condensed tannins was also descripted [89]. An example of this enzymatic activity was reported during a spontaneous fermentation of lentils [68]. Eight catechin and epicatechin dimers, trimers and tetramers were identified in raw lentils. Their content was halved after a spontaneous fermentation with a consequent increase, of the monomeric forms. A similar outcome, was observed during the fermentation of a soy product with *Bacillus pumilus* [55], nevertheless, lactobacilli, and bifidobacteria from gut microbiota as well as a strain of *Lb. plantarum* isolated from cheese, were found to cleave the heterocyclic ring of monomeric flavan-3-ols, giving rise to 1-(3′,4′-dihydroxyphenyl)-3-(2″,4″,6″-trihydroxyphenyl)propan-2-ol (3,4-diHPP-2-ol) [90]. This compound, which can be further degraded by the gut microbiota in several other substances, was recently studied for its antioxidant properties [91]. The authors found that 3,4-diHPP-2-ol had scavenging activity on ABTS and DPPH radicals higher than its precursor, as well as than other phenolic compounds. In addition, 3,4-diHPP-2-ol also showed the ability to reduce Fe^3+^ to Fe^2+^, higher than Trolox.

### 4.2. Release of Antioxidant Peptides

With the bioactive peptides crypted in the native protein, the active amino acid sequences need to be released through proteolytic microorganisms. Most of the studies involving the formation of bioactive peptides by fermentation are carried out by lactic acid bacteria which own a complex system of proteases and peptidases. Their proteolytic system consists of an extracellular proteinase, a transport system specific for small peptides, and a multitude of intracellular specific, generic, endo-, and eso-peptidases [92]. A pool of selected lactic acid bacteria (comprising *Lactobacillus alimentarius*, *Lb. brevis*, *Lb. sanfranciscensis*, and *Lactobacillus hilgardii*) was selected to ferment several cereal flours [32]. The highest antioxidant activity on DPPH and inhibition of linoleic acid autoxidation were found for fermented whole wheat, spelt, rye, and kamut. Twenty-five peptides (8-57 amino acid residues), identified by nano-liquid chromatography-electrospray ionization-tandem mass spectrometry (nano-LC-ESI-MS/MS), showed ex vivo antioxidant activity on mouse fibroblasts artificially subjected to oxidative stress [32]. Galli et al. [93] adopted the same experimental design selecting 23 out of 131 LAB strains to singly inoculate wheat flour doughs. The sourdough extracts reduced ROS formation in several cell lines (RAW 264.7 murine macrophage, murine H-end endothelium cells and Human intestinal Caco-2 cells). Another recent study exploited the effect of fermentation with *Lb. plantarum* and a commercial protease on lentils [67]. Due to the flavonoids liberated during the bioprocessing, a reduction of ROS on RAW 264.7 macrophage cell line was observed. Bioprocessed lentils were later subjected to simulated gastro-intestinal digestion and, since it was observed an increase in the inhibition of ROS formation, the authors suggested that the pH changes caused structure modifications of both phenolics and peptides, releasing even more bioactive sequences [67].

Twenty-seven selected LAB were also used to ferment quinoa flours and screened for the ability to improve antioxidant activity [33]. The highest scavenging activity on DPPH radical (more than 80%) was found when fermentation was carried out with *Lb. plantarum* T6B4, T6C16, and T0A10, whereas only *Lb. plantarum* T1B6, T6B4, and T0A10 showed antioxidant activity on ABTS radical. Since the last strain also enabled the highest inhibition of linoleic acid autoxidation, it was the only extract further subjected to the purification and characterization of antioxidant peptide fractions. Five peptides, with sizes from 5 to 9 amino acid residues, identified by nano-LC-ESI-MS/MS, showed antioxidant activity on human keratinocytes NCTC 2544 artificially subjected to oxidative stress and resulted resistant to further hydrolysis by digestive enzymes [33]. Ex vivo antioxidant properties of extracts of quinoa seed fermented with *Rhizopus oligosporus* were also evaluated by Matsuo [94]. The extracts increased both activities of superoxide dismutase and glutathione peroxidase ex vivo, therefore, they were used to feed rats for 13 days, confirming the results in vivo.

Several studies reported the ability of *Bacillus* sp., during soybean and wheat germ fermentation, to improve the in vitro antioxidant activity on DPPH and ABTS radicals. However, the activity was only ascribed to peptides; it was not demonstrated on purified fractions, nor were the sequences identified [95,96,97,98]. Conversely, rice proteins, residues from starch extraction, were hydrolyzed with a mixture of commercial proteolytic enzymes and *Bacillus pumilus* AG1, showing antioxidant activity towards ABTS radical. Almost all the peptides contained in the hydrolysate showed one or more features typical of well-known antioxidant peptides, most probably conferring a synergic antioxidant effect to the mixture with the potential to be used as functional ingredient [99].

Nevertheless, fungal eso- and endo-proteases play an important role in physiology and development of fungi, they are widely used in food industry [100], and examples of antioxidant peptides obtained by fungal fermentation were also reported (Table 2). Fermentation of soy flour with *Aspergillus oryzae* enabled the hydrolysis of native proteins into small molecular weight peptides, 90% of which were less than 3 kDa. The fermented soy was found to have antioxidant activity on DPPH radical higher than that of the positive controls (α-tocopherol and γ-oryzanol). It also inhibited 51.2% of linoleic acid oxidation, which was equivalent to 77% of the antioxidant activity of α-tocopherol [101].

As for phenolic compounds, several research papers focused on the antioxidant properties of soybean peptides as a result of fermentation with *Bacillus subtilis* [52,53], *Lb. plantarum* [102], *R. microspores* [103], *A. oryzae* [104], *Bifidobacterium* sp. [105]. Among the properties evaluated, the peptides possessed strong Cu^2+^ chelation ability, superoxide radical scavenging activity, reducing power potential and some of the peptides considered also resisted simulated gastro-intestinal digestion [53]. In one case, after assessing the radical scavenging activities on DPPH and ABTS, and the chelating ability of ferrous ions in vitro, the antioxidant properties of fermented soy were proven in vivo on rats, confirming the increase of superoxide dismutase activity in liver and kidney, and glutathione peroxidase activity in kidney [104]. A very recent study exploited the nutritional properties of a traditional soy-based Indonesian fermented food bought at local business with different level of sanitation [106]. The sanitation conditions clearly influence the microflora involved in the fermentation therefore affecting the formation of functional peptides. The identified peptides from the three different stores, differing in both number and molecular weight, showed similar and dissimilar features regarding amino acid sequences and functionalities. Yet, it was the cleanest production facility that had the highest number of peptides associated with functional properties including antioxidant activity [106]. Hence, this study enlightened the considerable impact of the starter in the success of the fermentation process.

### 4.3. Secondary Effects of Fermentation

#### 4.3.1. Vitamins

The ability of fermentation to modify phenolic and protein composition strictly relies on the metabolic activities of the specific starter used; however, the effect of acidification on endogenous proteinases or other enzymes involved in phenolics and protein metabolism cannot be excluded [69,92], as well as improving phenol solubility [49]. Few authors have also reported changes in vitamin content during fermentation [42,61,109] (Table 3). Vitamin E, also known as tocopherol, is an important antioxidant, which, due to a chroman group, halts lipid peroxidation by donating its phenolic hydrogen to peroxyl radicals forming tocopheroxyl radicals that are unreactive and unable to continue the oxidative chain reaction [1]. However, only photosynthetic microorganisms are known to accumulate detectable amounts of tocopherols; therefore, other factors participate to its increase during some food fermentations [110]. Fermented soy germ extracts exhibited a higher inhibition effect against the superoxide anion radical and lesser but significant ferric-reducing and DPPH radical scavenging effects compared to raw soy germ, which was ascribed to an increase in phenolic acids and isoflavones but also to tocopherols [61]. On the contrary, a decrease in tocopherol was observed in lupins fermentation, both spontaneous and inoculated with a strain of *Lb. plantarum* [111], Małgorzata et al. [112] instead, observed a decrease in the scavenging activity on ABTS of buckwheat fermented with *R. oligosporus*, despite the increase of tocopherol content. When *Cordyceps sinensis*, a fungus used in Chinese traditional medicine, was used to ferment stale rice, vitamin E concentration was doubled compared to unfermented rice and superoxide dismutase activity increased. Fermented rice was administrated to mice for 40 days inhibiting oxidative enzymes in brain and liver therefore delaying senescence [109]; and similar effects were also obtained by feeding rabbits with extracts of red bean fermented with *B. subtilis* [113]. To a combined action of polyphenols and vitamins was ascribed the protective effect against oxidative stress in mice after the consumption of bread made with Kamut sourdough [42,114]. However, Kamut breads, especially those with sourdough had a high content of selenium, which is known to participate in cells oxidative stress protection [114].

Great attention has gained a preparation known as Lisosan G^®^, a mineral- and vitamin-rich powder registered as nutritional supplement, which consists in wheat germ and bran fermented with a mix of lactobacilli and yeast strains [119]. Its antioxidant activity in vitro was confirmed after supplementation in the diet of 40 rabbits for 60 days. Lisosan G was found to reduce reactive oxygen metabolites and increase vitamin A and E concentrations in the blood. In addition, it caused the induction of antioxidant enzymes in the liver and kidney of the treated rabbits [117].

#### 4.3.2. Production of Exopolysaccharides

Exopolysaccharides (EPS) are long-chain polysaccharides produced by microorganisms, as a response to environmental stress, using various sugars as substrates. EPS are either associated with cell surfaces or secreted into the environment and can be classified as homo- or hetero-exopolysaccharides, depending on the composing sugar units [120]. The EPS biosynthetic pathway is very complex and includes several enzymes and glucose-6-phosphate appears to be a key intermediate linking between the anabolic pathways of EPS production and the catabolic pathways of sugar degradation [121]. EPS can be produced by bacteria, yeasts, or filamentous fungi and their physiological role depends on the microorganism producing them [120]. EPS are mostly employed in food industry as texture modifiers; however, fungal and bacterial EPS have been proved to have anticancer, antimicrobial, hypocholesterolemic, hypoglycemic, and also antioxidant activity [122]. It was suggested that their radical scavenging ability is associated to the molecular weight and the number of hydroxyl and amino groups, yet the relationship between antioxidant activity and physico-chemical properties or structural features is still uncertain due to opposite results [122]. Many purified EPS produced during fermentation with species of the genera *Lactobacillus* [123,124,125,126], *Lactococcus* [127,128], *Cordyceps* [129,130], *Aspergillus* [131] were found to have antioxidant activity in vitro and in vivo. Scavenging activity towards DPPH, ABTS and OH radicals, metal ion chelating ability, and inhibition of linoleic acid peroxidation, as well as protective effect against Caco-2 cells oxidative stress and increased superoxide dismutase in mice are among the properties demonstrated; however, there are few studies that have confirmed it during food fermentation. A study conducted few years ago, explored the potential of wheat distillers’ grains water extract fermentation to produce EPS by *Preussia aemulans* [116]. Compared to the unfermented extract, the fermented one had 36% of EPS higher and the scavenging activity against DPPH, ABTS, and OH radicals were assessed. One of the EPS fractions showed antioxidant activity comparable to that of ascorbic acid, as well as high ability to chelate metal ions and these properties were found to be dose-dependent [116]. Uchida et al. [115] studied the effect of rice kefiran, an EPS produced in a rice medium by *Lactobacillus kefiranofaciens*, on the diet of rabbits. The authors concluded that kefiran prevents the onset of atherosclerosis in hypercholesterolemic rabbits through antinflamatory and antioxidant actions [118].

## 5. Conclusions

The increasing interest in nutraceuticals reflects consumers’ attention towards studies indicating that specific diets or components in the diet are associated with lower risk of certain disease. Consequentially, consumer trends have shifted towards super foods that not only fulfill basic nutritious requirement, but also exert any number of functional features while being natural without additives. The major role of antioxidant compounds in preserving food shelf life, as well as providing health promoting benefits, combined with the increasing concern towards synthetic antioxidants, has led many authors to look for natural ways of increasing their content through fermentation. Phenolic compounds are the most studied substances displaying such properties, nevertheless, also peptides and protein derivatives, vitamins, and EPS, released or produced by the complex microbial enzymatic system, have been proven to exert antioxidant activities in vitro, ex vivo and in vivo (Figure 1).

Since most of the metabolic activities responsible for the antioxidant properties highlighted above are species- or strain- specific, starters selection is a pivotal step, and, even though the strains employed in these fermentations are recognized as safe, process parameters as well as many other factors, both affecting microbial and nutritional quality of fermented foods, need to be considered. Furthermore, as previously elucidated, an activity in vitro does not always correspond to an actual physiological function in vivo due to modifications occurring during the gastrointestinal digestion, therefore affecting the potential bioavailability of such compounds. Fermentation has the potential to meet consumers’ requirements, yet more in vivo studies are needed.

## Figures and Tables

**Figure 1 foods-08-00362-f001:**
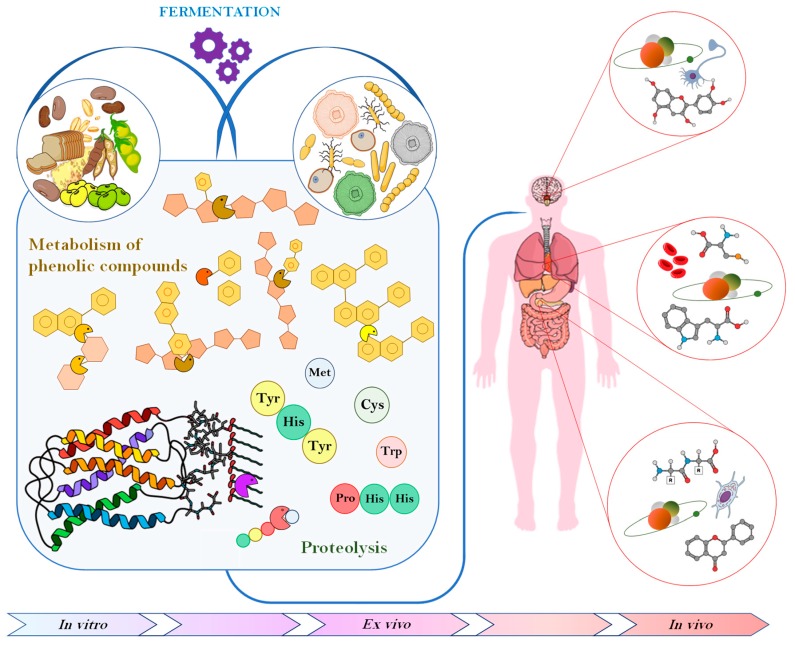
Schematic representation of the main effects of fermentation on food antioxidant compounds and consequent impact on human body.

**Table 1 foods-08-00362-t001:** Main effect of polyphenols metabolism on the antioxidant activity of fermented cereals and legumes.

Matrix	Microorganisms Employed	Process Parameters	Effect	Reference
Wheat bran	Baker’s yeast	20 °C for 20 h	Release of phenolic acids and improved bioaccessibility and colonic metabolism of phenolic acids	[38]
	Spontaneous fermentation conducted mainly by *Lactobacillus*, *Leuconostoc* and *Pediococcus* spp.	Backslopping for 13 days at 18 °C	Release of ferulic acid	[39]
	Baker’s yeast and LAB	20 °C for 24 h in anaerobic condition	Conversion of ferulic and caffeic acids into their derivatives and increase in sinapic acid.	[40]
	*Hericium erinaceus* and enzymes	25 °C for 7 days	Release of ferulic acid	[41]
Kamut bread	Baker’s yeast and spontaneously fermented sourdough	30 °C for 1.5 h	Response to oxidative stress in in vivo studies with rats.	[42]
Rice bran	*Rhizopus oryzae* CTT 1217	30 °C for 120 h	Increase ferulic acid and DPPH scavenging activity. Inhibition of peroxidase and polyphenol oxidase.	[43]
	*Rhizopus oligosporus* F0020 *Monascus purpureus* F0061	32 °C for 12 days	Release of phenolic acids and increase of FRAP	[44]
	*Bacillus subtilis* subsp. *subtilis* NRRL NRS-744	37 °C for 8 days	Release of phenolic acids	[45]
Rye bran	Baker’s yeast	25 °C for 14 h	Increase of phenolic acids	[46]
	Baker’s yeast	20 °C for 24 h	Release of phenolic acids but no improved bioaccessibility in in vitro digestive systems	[47,48]
Sorghum flour	*Lactobacillus plantarum* FUA3171, *Lactobacillus casei* FUA3166, *Lactobacillus fermentum* FUA3165 *Lactobacillus reuteri* FUA3168	34 °C for 24 h	Release of phenolic acids and flavonoids	[49]
Tef pancake	Spontaneously fermented sourdough	25 °C for 24-120 h	Solubilization of bound phenolics and improved antioxidant potentials on FRAP and ABTS	[50]
Malt based beverage	*Lb. plantarum* Lp758, Lp765, Lp725, *Lactobacillus brevis* Lb986, *Lactobacillus amylolyticus* LaTL3, LaT15	30 °C for 72 h	Decarboxylation of phenolic acids.	[51]
Soy	*Bacillus subtilis* SHZ	30 °C for 36 h	Superoxide radical scavenging activity and reducing power potential	[52]
	*B. subtilis* MTCC5480, MTCC1747	42 °C for 24 h	Increase in the antioxidant activity due to both phenolic compounds and peptides	[53]
	*Aspergillus oryzae* ATCC 1011 *Rhizopus oryzae* ATCC 24563 *Bacillus subtilis* ATCC 6051	30 °C for 48 h	Increase in phenolic acids and flavonoids	[54]
	*Bacillus pumilus* HY1	37 °C for 60 h	Increase in phenolic acids, flavonoids and tannins monomeric forms	[55]
	*Aspergillus awamori* *Aspergillus niger* *Aspergillus niveus*	30 °C for 5 days	Increase in phenolic acids and flavonoids as consequence of β-glucosidase activity	[56]
	*R. oligosporus*	Room temperature for 20 days	Increase in phenolic acids and flavonoids	[57]
	*Lentinus edodes* CY-35	Room temperature for 50 days	Increase in phenolic acids and flavonoids as consequence of laccase and β-glucosidase activities	[58]
	*Bacillus natto*	37 °C for 48 h	Increase in phenolic acids and flavonoids	[59]
	*Lactobacillus plantarum* CECT 748 T	30 °C for 48 h	Increase in phenolic acids and flavonoids	[60]
	Pool of selected LAB	37 °C for 48 h	Increase in phenolic acids, flavonoids, saponins, phytosterols, and tocopherols	[61]
	*Lactobacillus casei* *Lactobacillus acidophilus*	32 °C for 15 h	Increased of the aglycones/glycosylated isoflavones ratio and bound phenolics	[62]
	*Lactobacillus casei* 0979 after germination	30 °C for 24 h	Increase in isoflavones glycosides and aglycones	[63]
	Kefir grains containing LAB and yeasts	30 °C for 24 h	Increase of isoflavones and improved antioxidant activities on DPPH and ABTS	[64]
	*Lb. plantarum* DPPMA24W, DPPMASL33 *Lb. fermentum* DPPMA114 *Lactobacillus rhamnosus* DPPMAAZ1	30 °C for 96 h	Increase of isoflavone aglycones especially equol	[65]
Cowpeas	Spontaneously fermented *Lb. plantarum* ATCC 14917	37 °C for 48 h	Increase of phenolic acids derivatives and flavonoids. Improved antioxidant activity on DPPH	[66]
Lentils	*Lb. plantarum* CECT 748 and commercial protease	37 °C for 15 h	Reduction of ROS on RAW 264.7 cells	[67]
	Spontaneously fermented	35 °C for 4 days	Decrease of condensed tannin and increase of monomers.	[68]

**Table 2 foods-08-00362-t002:** Main effect of bioactive peptides and amino acids derivatives on the antioxidant activity of fermented cereals, pseudocereals, and legumes.

Matrix	Microorganisms Employed	Process Parameters	Effect	Reference
Wheat flour	*Lactobacillus farciminis* A11, A19, H3, *Lactobacillus rossiae* A20, Gd40, *Lactobacillus sanfranciscensis* B3, I4, *Lactobacillus plantarum* O4, *Lactobacillus brevis* A7	30 °C for 24 h	Reduction of ROS on RAW 264.7, H-end and Caco-2 cells	[93]
Defatted wheat germ	*Bacillus subtilis* B1	37 °C for 24 h	Unidentified peptides	[96]
Rice protein	*Bacillus pumilus* AG1	37 °C for 72 h	Identified peptide sequences with high antioxidant activity	[99]
Wheat, spelt, rye, and kamut flours	Pool of selected LAB	37 °C for 24 h	Identified peptide sequences with high antioxidant activity	[32]
Quinoa	*Rhizopus oligosporus* NRRL2710	36 °C for 24 h	Increased ex vivo and in vivo activities of superoxide dismutase, GSHPx, and TBARS	[94]
	*L. plantarum* T0A10	37 °C for 24 h	Identified peptides with antioxidant activity on human keratinocytes NCTC 2544	[33]
	*R. oligosporus* ATCC 64063 *Aspergillus oryzae* DSM 1861 *Neurospora intermedia* DSM 1965	31 °C for 6 days 25 °C for 6 days 30 °C for 5 days	Increase of OH and ABTS radical scavenging activity due to potentially bioactive peptides	[107]
Soy	Spontaneously sourdough containing *Rhizopus* spp., LAB and yeasts	Room temperature for 2 days	Identified peptide sequences with potential antioxidant activity	[106]
	*B. subtilis* SHZ	30 °C for 36 h	Superoxide radical scavenging activity and reducing power potential	[52]
	*B. subtilis* MTCC5480, MTCC1747	42 °C for 24 h	Increase in the antioxidant activity due to both phenolic compounds and peptides	[53]
	*Lb, plantarum* Lp6	30 °C for 24 h	Identified peptide sequences with high antioxidant activity	[102]
	*Aspergillus orizae*	30 °C for 3 days 45 °C for 4 days	Unidentified low molecular weight peptides	[100]
	*Rhizopus microsporus*	36 °C for 25 h	Improvement in the antioxidant activity attributed to amino acids and peptides	[97]
	*Bifidobacterium* sp.	37 °C for 48 h	Inhibition of ascorbate autoxidation, superoxide radical scavenging activity and reducing power potential peroxide	[105]
	*Aspergillus oryzae*	30 °C for 60 h	Superoxide dismutase and glutathione peroxidase activities *in vivo*	[104]
Kidney beans	*B. subtilis* CECT 39T *Lb. plantarum* CECT 748T	30 °C for 96 h 37 °C for 96 h	Improved antioxidant activity on ORAC-FL	[108]
Lentils	*Lb. plantarum* CECT 748 and commercial protease	37 °C for 15 h	Reduction of ROS on RAW 264.7 cells	[67]

**Table 3 foods-08-00362-t003:** Secondary effect of fermentation on the antioxidant activity of cereals and legumes.

Matrix	Microorganisms Employed	Process Parameters	Effect	Reference
Stale rice	*Cordyceps sinensis*	25 °C for 7 days	Increase of tocopherol and superoxide dismutase activity in mice	[109]
Rice medium	*Lactobacillus kefiranofaciens*		EPS responsible of atherosclerosis prevention	[115]
Wheat distillers’ grains	*Preussia aemulans*	Room temperature for 7 days	EPS formation with high radical scavenging activity and metal ions chelating ability	[116]
Wheat germ and bran	Yeasts and lactobacilli		High vitamin content responsible for in vivo antioxidant activity in liver and kidney	[117]
Kidney beans	*Bacillus subtilis* IMR-NK1	30 °C for 48 h	Improved antioxidant activity on DPPH, reducing power potential and Fe^2+^ chelating ability	[118]
	*Bacillus subtilis* BCRC 14716	30 °C for 48 h	Increased vitamin E levels in liver and brain of rabbit, and superoxide dismutase activity in the brain	[113]

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
