# Peer review of "How Fermentation Affects the Antioxidant Properties of Cereals and Legumes"

_foods, 2019, doi:10.3390/foods8090362_

Round 1

Reviewer 1 Report

The manuscript entitled “How fermentation affects the antioxidant properties of foods” is a review of food fermentation and the antioxidant capacity associated with grains.  The review is set-up in manner that is logical and flows between each section, but also allows for quick identification of the pertinent required section when scanning the review for relevant information on a single aspect.  The review does address a sector of the food processing industry that is often overlooked.  The manuscript will be of interest to readers involved in antioxidant research as well as those in the food sector looking to improve antioxidant activity without the use of synthetic antioxidants.

The manuscript does require English editing and there are several aspects that need clarified, prior to publication.

General

Ensure all abbreviations are defined in the first instance.

Abstract

Line 15 ‘present’

Keywords:

Additional keywords should be supplied to help the researcher find your manuscript. ‘grains’ would be one that should be included.

Section 1

The authors highlight briefly their selection process for the articles evaluated. I think it would be useful for the to specify the search engine they used and the results associated with each criteria.  For example: X number of articles on fermented foods found in PubMed,  of these X number quantify antioxidant activity, and of these X number utilize HPLC in conjunction with an antioxidant capacity assay, which is our researched population.

Line 45-48. The section is unclear as to what the authors are trying convey. 

Line 60: Provide examples of what these other protein derived compounds are.

Line 76-78: What is the relationship between diets in developing countries and Western vegetarian diets?  This seems like it is two thoughts randomly merged together, please clarify.

Section 1.1

Line 85-86. Reword the sentence for clarity.

Line 86: ‘Phenolics are composed of …’

Line 88. ‘… structural elements that are bound to the ring.’

Line 104-107.  These are important facts and should be referenced.

Section 1.2

Line 111. ‘… responsible for health …’

Line 114. ‘…, storage, or in vitro…’

Line 123 ‘ proven …’

Line 133. The authors speak of a correct sequence for peptide activity.  What is the correct sequence?

Section 2.

Line 156. Remove ‘ because’.

Line 159-160.  The sentence is unclear.

Line 165. Omit ‘to’.

Line 166. ‘… they may or may not still exert their …’

Line 172 -173.  After the word plasma the sentence is unclear.

Line 177-178.  The authors appear to be putting personal bias into the argument that in vivostudies are ‘tedious’ and ‘require ethics approval’.  This is not a strong argument to not get approval and perform these studies. 

Section 3.1

Line 188. ‘… to assess food …’

Line 189. What is ‘They’ referring to the assays or antioxidants?

Line 189. What are the MOA categories? Traditionally they are thought of as hydrogen atom transfer and electron transfer assays. 

Section 3.2

Line 203. ‘animals’

Line 224. ‘… major sources responsible …’

Section 3.3

Line 230. Hydroperoxides and its measurements are considered what?  Complete the thought.

Section 4.1.1

Line 244-245.  ‘… cytoplasm or bound’  but bound to what?  Complete the thought.

Line 250 -251.  What are the paths that the strains follow?

Line 269. ‘… acids into their …’

Line 271. ‘ferulic acid’

Line 273. ‘bound’

Section 4.1.3

Line 349-351.  The sentence is unclear as to the idea that the authors are conveying.

Section 4.2

Line 392. ‘… fractions, nor were the sequences identified.’

Line 401. ‘… which are less …’

References

Please ensure all references meet the Guidelines for authors.  There are several instances where the year is not bolded (lines 819, 823, and others).

The authors have very few references in the 3 and 5 years.  A limited search in this time frame produces several articles that are relevant to this manuscript.  i.e. Frontiers in Nutrition doi: 10.3389/fnut.2019.00042 and Molecules 24(14) 2606, both from 2019.

Tables 1, 2, 3

The tables are not referenced in the text.  Tables and figures should support the discussion in the text and be referenced to there support.  

Figure 2.

Figure 2 does not seem linked to this conclusion statement as written.  The figure would be more appropriate as a graphical abstract.  However, it would need to be modified as the review is about antioxidant properties after fermentation, not the implications within the human after digestion, which is what the figure implies.

Author Response

The manuscript entitled “How fermentation affects the antioxidant properties of foods” is a review of food fermentation and the antioxidant capacity associated with grains.  The review is set-up in manner that is logical and flows between each section, but also allows for quick identification of the pertinent required section when scanning the review for relevant information on a single aspect.  The review does address a sector of the food processing industry that is often overlooked.  The manuscript will be of interest to readers involved in antioxidant research as well as those in the food sector looking to improve antioxidant activity without the use of synthetic antioxidants.

The manuscript does require English editing and there are several aspects that need clarified, prior to publication.

General

Ensure all abbreviations are defined in the first instance.

Done

Abstract

Line 15 ‘present’

Ok.

Keywords:

Additional keywords should be supplied to help the researcher find your manuscript. ‘grains’ would be one that should be included.

Ok.

Section 1

The authors highlight briefly their selection process for the articles evaluated. I think it would be useful for the to specify the search engine they used and the results associated with each criteria.  For example: X number of articles on fermented foods found in PubMed,  of these X number quantify antioxidant activity, and of these X number utilize HPLC in conjunction with an antioxidant capacity assay, which is our researched population.

Thank you for the suggestion. The selection process has been better explained.

Line 45-48. The section is unclear as to what the authors are trying convey.

The sentence has been clarified.

Line 60: Provide examples of what these other protein derived compounds are.

A short description has been added (line 59)

Line 76-78: What is the relationship between diets in developing countries and Western vegetarian diets?  This seems like it is two thoughts randomly merged together, please clarify.

It was not intended to highlight a relationship between the diets in developing countries and Western vegetarian diets. We tried to emphasize the importance of legumes and cereals in both diets. To better clarify this concept, the sentence has been rewritten.

Section 1.1

Line 85-86. Reword the sentence for clarity.

The sentence has been modified.

Line 86: ‘Phenolics are composed of …’

Ok.

Line 88. ‘… structural elements that are bound to the ring.’

Ok.

Line 104-107.  These are important facts and should be referenced.

References have been added.

Section 1.2

Line 111. ‘… responsible for health …’

Ok.

Line 114. ‘…, storage, or in vitro…’

Ok.

Line 123 ‘ proven …’

Ok.

Line 133. The authors speak of a correct sequence for peptide activity.  What is the correct sequence?

Ok. ‘Correct positioning in the sequence’ was used to indicate the dependence of to antioxidant activity to the different position of some aminoacids in the peptide sequence. The sentence has been modified.

Section 2.

Line 156. Remove ‘ because’.

Ok.

Line 159-160.  The sentence is unclear.

Depending on the level of polymerization or glycosylation, the other phenolic compounds here mentioned can have higher molecular weight. Therefore, although smaller compounds like phenolic acids are the ones easily absorbed, such property was reported for the others as well. The sentence has been modified.

Line 165. Omit ‘to’.

Ok.

Line 166. ‘… they may or may not still exert their …’

Ok.

Line 172 -173.  After the word plasma the sentence is unclear.

Yes. The sentence has been rewritten.

Line 177-178.  The authors appear to be putting personal bias into the argument that in vivostudies are ‘tedious’ and ‘require ethics approval’.  This is not a strong argument to not get approval and perform these studies.

The authors agree with the reviewer. The sentence has been deleted and the issues related to the in vivo studies better clarified.

Section 3.1

Line 188. ‘… to assess food …’

Ok.

Line 189. What is ‘They’ referring to the assays or antioxidants?

‘They’ refers to the methods. The sentence has been modified.

Line 189. What are the MOA categories? Traditionally they are thought of as hydrogen atom transfer and electron transfer assays.

Antioxidants mechanisms of action are described in section 1 (lines 50-54). However, the sentence has been modified.

Section 3.2

Line 203. ‘animals’

Ok.

Line 224. ‘… major sources responsible …’

Ok.

Section 3.3

Line 230. Hydroperoxides and its measurements are considered what?  Complete the thought.

The sentence has been rewritten.

Section 4.1.1

Line 244-245.  ‘… cytoplasm or bound’  but bound to what?  Complete the thought.

The sentence has been completed.

Line 250 -251.  What are the paths that the strains follow?

Phenolic acids can be either decarboxylated or reduced by lactic acid bacteria, the concept has been clarified.

Line 269. ‘… acids into their …’

Ok.

Line 271. ‘ferulic acid’

Ok.

Line 273. ‘bound’

Ok.

Section 4.1.3

Line 349-351.  The sentence is unclear as to the idea that the authors are conveying.

The sentence has been clarified.

Section 4.2

Line 392. ‘… fractions, nor were the sequences identified.’

Ok.

Line 401. ‘… which are less …’

Ok.

References

Please ensure all references meet the Guidelines for authors.  There are several instances where the year is not bolded (lines 819, 823, and others).

The references have been checked.

The authors have very few references in the 3 and 5 years.  A limited search in this time frame produces several articles that are relevant to this manuscript.  i.e. Frontiers in Nutrition doi: 10.3389/fnut.2019.00042 and Molecules 24(14) 2606, both from 2019.

The authors thank the reviewer for the suggestion. The direct reference to the first article you suggest has been added (we also checked the papers listed in this article).

However, the manuscript published in Molecules (the second you suggest) does not fit the selection criteria, since the antioxidant activity was performed on the strains and not on the product (kefir).

The revised version of the manuscript includes, with the exception of few papers, references to research papers published from 2008 onwards (half of which in the last 5 years).

Tables 1, 2, 3

The tables are not referenced in the text.  Tables and figures should support the discussion in the text and be referenced to there support. 

Table have been cited in the text.

Figure 2.

Figure 2 does not seem linked to this conclusion statement as written.  The figure would be more appropriate as a graphical abstract.  However, it would need to be modified as the review is about antioxidant properties after fermentation, not the implications within the human after digestion, which is what the figure implies.

We believe that the interest, of both scientific and public communities, in antioxidant compounds is mainly due to their effect on the human body. Since many studies involving cereals and legumes fermentation demonstrated an improved antioxidant activity both ex vivo and in vivo, the authors believe illustrating the implications within the body after digestion is an important part of the general picture this review is trying to represent and is in line with the conclusions. We would like to keep this figure in the final version of the manuscript.

Reviewer 2 Report

This is very comprehensive review on the effect of fermentation on the antioxidant properties of select foods (cereals and legumes). The authors summarize the main antioxidant components in food matrices, the bioaccessibility and bioavailability of antioxidant compounds, different antioxidant assays (in vitro, ex vivo and in vivo) and finally the effect of fermentation on the antioxidant components in different food materials. Overall this is a well-written, well-organized review focusing on the potential of fermentation on increasing the antioxidant properties of foods.

Some suggestions:

Title: this review specifically focused on cereals and legumes, therefore the title should be more specific. About the classification of condensed tannins (proanthocyanidins). Proanthocyanidins are oligomers and polymers of flavan-3-ols and are normally considered as part of the flavonoid compound family. Consider adding more discussion on the limitation of different antioxidant assays mentioned in section 3. Some of these assays were used to evaluate the antioxidant properties of fermented foods in sections 4’s studies, readers should be aware of their limitations. A table summarizing their mechanisms, accuracy, and limitations etc. can be added.   

Reviewer 3 Report

The study presents the impact of fermentation on the antioxidant properties of foods. After a careful survey, I came to the conclusion that some points need to be further explained or revised.

Specific comments:

The sections 1. Phenolic compounds, 1.2. Antioxidant peptides and protein derivatives, 1.3. Synthetic antioxidants contain generalities and could be shortened. In my opinion structural formulas of described compounds (Fig. 1, Phenolic compounds classification) are generally known and unnecessary in this publication. Section References is too long and some positions should be removed.

Author Response

The study presents the impact of fermentation on the antioxidant properties of foods. After a careful survey, I came to the conclusion that some points need to be further explained or revised.

Specific comments:

The sections 1. Phenolic compounds, 1.2. Antioxidant peptides and protein derivatives, 1.3. Synthetic antioxidants contain generalities and could be shortened. In my opinion structural formulas of described compounds (Fig. 1, Phenolic compounds classification) are generally known and unnecessary in this publication. Section References is too long and some positions should be removed.

The figure representing the classification of phenolic compounds was removed. Of the 132 references in the list, almost a hundred strictly refer to the metabolic pathways of microbial species on phenolic compounds and the effect of fermentation on antioxidant activity. The rest of the references were used for the first three section of the review. However, where possible, unnecessary concepts in the sections indicated have been removed.

Reviewer 4 Report

This review aims at providing a comprehensive overview of the effect of fermentation on the antioxidant activity of vegetable matrices.

The major role of antioxidant compounds in preserving food shelf life, as well as providing health promoting benefits, combined with the increasing concern towards synthetic antioxidants, has led many authors to look for natural ways of increasing their content through fermentation. 

There is nice and magnificent work to produce this review.

Author Response

This review aims at providing a comprehensive overview of the effect of fermentation on the antioxidant activity of vegetable matrices.

The major role of antioxidant compounds in preserving food shelf life, as well as providing health promoting benefits, combined with the increasing concern towards synthetic antioxidants, has led many authors to look for natural ways of increasing their content through fermentation.

There is nice and magnificent work to produce this review.

The authors thank the reviewer.